# Linguistic Binding in Diffusion Models: Enhancing Attribute Correspondence through Attention Map Alignment

**Royi Rassin**
Bar-Ilan University, Israel
`rassinroyi@gmail.com`

**Eran Hirsch**
Bar-Ilan University, Israel
`eran.hirsch@biu.ac.il`

**Daniel Glickman**
Bar-Ilan University, Israel
`danielglickman1@gmail.com`

**Shauli Ravfogel**
Bar-Ilan University, Israel
Allen Institute for AI, Israel
`shauli.ravfogel@gmail.com`

**Yoav Goldberg**
Bar-Ilan University, Israel
Allen Institute for AI, Israel
`yoav.goldberg@gmail.com`

**Gal Chechik**
Bar-Ilan University, Israel
NVIDIA, Israel
`gal.chechik@biu.ac.il`

## Abstract

Text-conditioned image generation models often generate incorrect associations between entities and their visual attributes. This reflects an impaired mapping between *linguistic binding* of entities and modifiers in the prompt and *visual binding* of the corresponding elements in the generated image. As one example, a query like "a *pink sunflower* and a *yellow flamingo*" may incorrectly produce an image of a *yellow sunflower* and a *pink flamingo*. To remedy this issue, we propose *SynGen*, an approach which first syntactically analyses the prompt to identify entities and their modifiers, and then uses a novel loss function that encourages the cross-attention maps to agree with the linguistic binding reflected by the syntax. Specifically, we encourage large overlap between attention maps of entities and their modifiers, and small overlap with other entities and modifier words. The loss is optimized during inference, without retraining or fine-tuning the model. Human evaluation on three datasets, including one new and challenging set, demonstrate significant improvements of SynGen compared with current state of the art methods. This work highlights how making use of sentence structure during inference can efficiently and substantially improve the faithfulness of text-to-image generation.[1]

## 1 Introduction

Diffusion models for text-conditioned image generation produce impressive realistic images [1, 2, 3, 4]. Users control the generated content through natural-language text prompts that can be rich and complex. Unfortunately, in many cases the generated images are not faithful to the text prompt [5, 6]. Specifically, one very common failure mode results from **improper binding**, where *modifier* words fail to influence the visual attributes of the *entity-nouns* to which they are grammatically related.

---

[1]We make our code publicly available `https://github.com/RoyiRa/Syntax-Guided-Generation`

37th Conference on Neural Information Processing Systems (NeurIPS 2023).

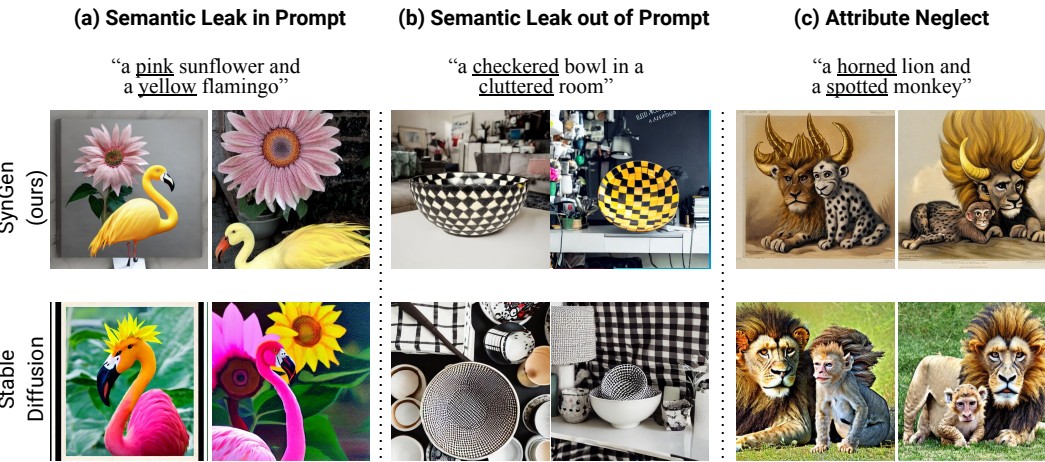

Figure 1: Visual bindings of objects and their attributes may fail to match the linguistic bindings between entities and their modifiers. Our approach, SynGen, corrects these errors by matching the cross-attention maps of entities and their modifiers.

As an illustration, consider the prompt "a *pink sunflower* and a *yellow flamingo*". Given this prompt, current models often confuse the modifiers of the two entity-nouns, and generate an image of a *yellow* sunflower and a *pink* flamingo (Fig. 1, bottom left, semantic leak in prompt). In other cases, the attribute may semantically leak to areas in the image that are not even mentioned in the prompt (Fig. 1, bottom center, semantic leak outside prompt) or the attribute may be completely neglected and missed from the generated image (Fig. 1, bottom right, attribute neglect). Such mismatch can be addressed by providing non-textual control like visual examples [7, 8], but the problem of correctly controlling generated images using text remains open.

A possible reason for these failures is that diffusion models use text encoders like CLIP [9], which are known to fail to encode linguistic structures [10]. This makes the diffusion process "blind" to the linguistic bindings, and as a result, generate objects that do not match their attributes. Building on this intuition, we propose to make the generation process aware of the linguistic structure of the prompt. Specifically, we suggest to intervene with the generation process by steering the cross-attention maps of the diffusion model. These cross-attention map serve as a link between prompt terms and the set of image pixels that correspond to these terms. Our linguistics-based approach therefore aims to generate an image where the **visual binding** between objects and their visual attributes adheres to the **syntactic binding** between entity-nouns and their modifiers in the prompt.

Several previous work devised solutions to improve the relations between prompt terms and visual components, with some success [11, 12, 13]. They did not focus on the problem of modifier-entity binding. Our approach specifically addresses this issue, by constructing a novel loss function that quantifies the distance between the attention patterns of grammatically-related (modifier, entity-noun) pairs, and the distance between pairs of unrelated words in the prompt. We then optimize the latent denoised image in the direction that separates the attention map of a given modifier from unrelated tokens and bring it closer to its grammatically-related noun. We show that by intervening in the latent code, we markedly improve the pairing between attributes and objects in the generated image while at the same time not compromising the quality of the generated image.

We evaluate our method on three datasets. (1) For a natural-language setting, we use the natural compositional prompts in the ABC-6K benchmark [13]; (2) To provide direct comparison with previous state-of-the-art in [11], we replicate prompts from their setting; (3) Finally, to evaluate binding in a challenging setting, we design a set of prompts that includes a variety of modifiers and entity-nouns. On all datasets, we find that SynGen shows significant improvement in performance based on human evaluation, sometimes doubling the accuracy. Overall, our work highlights the effectiveness of incorporating linguistic information into text-conditioned image generation models and demonstrates a promising direction for future research in this area.

Figure 2: The SynGen workflow and architecture. (a) The text prompt is analyzed to extract entity-nouns and their modifiers. (b) SynGen adds intermediates steps to the diffusion denoising process. In that step, we update the latent representation to minimize a loss over the cross attention maps of entity-nouns and their modifiers (Eq 3).

The main contributions of this paper are as follows: (1) A novel method to enrich the diffusion process with syntactic information, using inference-time optimization with a loss over cross-attention maps; (2) A new challenge set of prompts containing a rich number and types of modifiers and entities.

## 2 Syntax-Guided Generation

Our approach, which we call SynGen, builds on two key ideas. First, it is easy to analyze the syntactic structure of natural language prompts to identify bindings of entity-nouns and their modifiers. Second, one can steer the generation of images to adhere to these bindings by designing an appropriate loss over the cross-attention maps of the diffusion model. We describe the two steps of our approach: extracting syntactic bindings and then using them to control generation.

### 2.1 Identifying entity-nouns and their modifiers

To identify entity-nouns and their corresponding modifiers, we traverse the syntactic dependency graph, which defines the syntactic relation between words in the sentence. Concretely, we parse the prompt using spaCy's transformer-based dependency parser [14] and identify all entity-nouns (either proper-nouns or common-nouns) that are not serving as direct modifiers of other nouns.

These are the nouns that correspond to objects in the generated image. We then recursively collect all modifiers[2] of the noun into its modifier set. The set of modifier-labels includes a range of syntactic relations between nouns and their modifiers, such adjectival modification (amod; "the *regal* dog"), compounds (compound; "the *treasure* map"), nominal modification through an intervening marker, adverbial modifiers (npadvmod; "A *watermelon*-styled chair"), adjectival complement (acomp; "The apple is *blue*"), and coordination between modifiers (conj; "A *black and white* dog").

### 2.2 Controlling generation with language-driven cross-attention losses

Consider a pair of a noun and its modifier. We expect the cross-attention map of the modifier to largely overlap with the cross-attention map of the noun, while remaining largely disjoint with the maps corresponding to other nouns and modifiers. To encourage the denoising process to obey these spatial relations between the attention maps, we design a loss that operates on all cross-attention maps. We then use this loss with a pretrained diffusion model during inference. Specifically, we

---

[2]We consider modifiers from the set {amod, nmod, compound, npadvmod, acomp, conj}. We exclude conj when determining the top-level nouns.

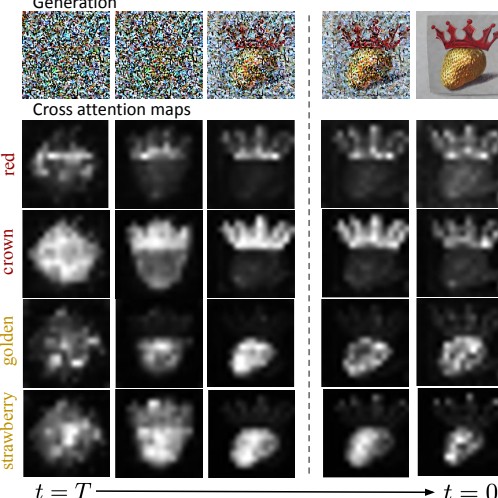

Figure 3: Evolution of cross-attention maps and latent representation along denoising steps, for the prompt "a red crown and a golden strawberry". At first, the attention maps of all modifiers and entity-nouns are intertwined, regardless of the expected binding. During denoising, attention maps gradually becomes separated, adhering the syntactic bindings. The vertical line indicates that after 25 steps intervention stops, but the attention maps remain separated.

optimize the *noised latents* by taking a gradient step to reduce that loss. See illustration in Fig. 2. Fig. 3 illustrates the effect of the loss over the cross-attention maps.

**Loss functions:** Consider a text prompt with $N$ tokens, for which our analysis extracted $k$ noun-modifier sets $\{S_1, S_2, \ldots, S_k\}$. Let $P(S_i)$ represent all pairs $(m, n)$ of tokens between the noun root $n$ and its modifier descendants $m$ in the $i$-th set $S_i$. For illustration, the set of "A black striped dog" contains two pairs ("black", "dog") and ("striped", "dog"). Next, denote by $\{A_1, A_2, \ldots, A_N\}$ the attention maps of all $N$ tokens in the prompt, and denote by $dist(A_m, A_n)$ a measure of distance (lack of overlap) between attention maps $A_m$ and $A_n$.

Our first loss aims to minimize that distance (maximize the overlap) over all pairs of modifiers and their corresponding entity-nouns $(m, n)$,

$$\mathcal{L}_{pos}(A, S) = \sum_{i=1}^{k} \sum_{(m,n) \in P(S_i)} dist(A_m, A_n). \tag{1}$$

We also construct a loss that compares pairs of modifiers and entity-nouns with the remaining words in the prompt, which are grammatically unrelated to these pairs. In other words, this loss is defined between words within the (modifiers, entity-nouns) set and words outside of it. Formally, let $U(S_i)$ represent the set of unmatched words obtained by excluding the words in $S_i$ from the full set of words and $A_u$ is the corresponding attention map for a given unrelated word $u$. The following loss encourages moving apart grammatically-unrealted pairs of words:

$$\mathcal{L}_{neg} = -\sum_{i=1}^{k} \frac{1}{|U(S_i)|} \sum_{(m,n) \in P(S_i)} \sum_{u \in U(S_i)} \frac{1}{2} \left( dist(A_m, A_u) + dist(A_u, A_n) \right). \tag{2}$$

Our final loss combines the two loss terms:

$$\mathcal{L} = \mathcal{L}_{pos} + \mathcal{L}_{neg}. \tag{3}$$

For a measure of distance between attention maps we use a symmetric Kullback-Leibler divergence $dist(A_i, A_j) = \frac{1}{2} D_{KL}(A_i || A_j) + \frac{1}{2} D_{KL}(A_j || A_i)$, where $A_i$, $A_j$ are attention maps normalized to a sum of 1, $i$ and $j$ are generic indices, and $D_{KL}(A_i || A_j) = \sum_{pixels} A_i \log(A_i / A_j)$.

Our test-time optimization approach resembles the one of [11], which defined a loss over the cross-attention maps to update the latents at generation time. However, their loss aims to maximize the presence of the smallest attention map at a given timestep to guarantee a set of selected tokens is included in the generated image, and our loss depends on pairwise relations of linguistically-related words and aims to align the diffusion process to the linguistic-structure of the prompt.

## 2.3 The workflow

We use the loss of Eqs 1-3 to intervene in the first 25 out of 50 denoising steps. Empirically, using a smaller number of steps did not correct well improper binding, and using a larger number generated blurred images, as detailed in Appendix B. In each of the first 25 steps, a pretrained denoiser (U-Net) was first used to denoise the latent variable $z_t$. Then, we obtained the cross-attention maps as in [15]. Next, we used the loss $\mathcal{L}$ to update the latent representation $z_t$ with a gradient step $z_t' = z_t - \alpha \cdot \nabla_{z_t} \mathcal{L}$. Finally, the U-Net architecture denoises the updated latent variable $z_t'$ for the next timestep.

## 3 Experiments

### 3.1 Compared baseline methods

We compare SynGen with three baseline methods. (1) Stable Diffusion 1.4 (SD) [1]; (2) Structured Diffusion [13], extracts noun-phrases from the prompt and embeds them separately, to improve the mapping of the semantics in the cross-attention maps; and (3) Attend-and-Excite (A&E) [11], a method that given a predetermined set of tokens, updates the latent a certain number of timesteps, to eventually incorporate these tokens in the generated image. To automate token selection in A&E, we follow the recommendation by the authors to select the nouns using a part-of-speech tagger.

### 3.2 Datasets

We evaluate our approach using two existing benchmark datasets, and one new dataset that we designed to challenge methods in this area.

**(1) ABC-6K [13].** This benchmark consists of 3.2K natural compositional prompts from MSCOCO [16], which were manually written by humans, using natural language and contain at least two color words modifying different noun-entities. In addition, the dataset contains 3.2K counterparts, where the position of modifiers in the original prompts are swapped. (e.g., "a *white* bench in front of a *green* bush" and "a *green* bench in front of a *white* bush"). We randomly sample 600 prompts.

**(2) Data from Attend-and-Excite [11].** Originally introduced to evaluate the A&E method which focuses on entity-neglect, this dataset also showed that A&E improved over previous work in terms of improper binding.

Prompts in this dataset belong to three categories: (1) "a {color} {in-animate object} and a {color} {in-animate object}"; (2) "a {color} {in-animate object} and an {animal}"; (3) "an {animal} and an {animal}". Following the split in A&E, we sample 33 prompts from type (1) and 144 prompts from type (2), but exclude type (3), as it does not contain modifiers. This is a very simple dataset, which we use to facilitate direct comparison with previous work.

**(3) Diverse Visual Modifier Prompts (DVMP).** The above two datasets are limited in terms of number and types of modifiers, and the number of entity-nouns per prompt. To challenge our model, we design a dataset consisting of coordination sentences, in similar fashion to the dataset from A&E, but with strong emphasis on the number and types of modifiers per prompt. Specifically, we aim to compare the models with prompts that contain numerous and uncommon modifiers, creating sentences that would not usually be found in natural language or training data, such as "a *pink spotted* panda". DVMP was designed with two key aspects in mind:

**Expanding the set of modifiers:** We have extended the number of modifiers referring to an entity-noun from one to up to three. For instance, "a *blue furry spotted* bird". We also added types of modifiers besides colors, including material patterns ("a *metal* chair"), design patterns ("a *checkered* shoe"), and even nouns modifying other noun-entities ("a *baby* zebra").

**Visually verifiable and semantically coherent:** The modifiers selected for DVMP are visually verifiable, with a deliberate avoidance of nuanced modifiers. For instance, "big" is a relative modifier dependent on its spatial context, and emotional states, such as in the prompt "an *excited* dog", are largely excluded due to their subjective visual interpretation. Simultaneously, DVMP maintains semantic coherence by appropriately matching modifiers to noun-entities, thereby preventing the creation of nonsensical prompts like "a sliced bowl" or "a curved zebra".

In total, we have generated 600 prompts through random sampling. For a comprehensive description of the dataset's creation, see Appendix F.

### 3.3 Human Evaluation

We evaluate image quality using Amazon Mechanical Turk (AMT). Raters were provided with a multiple-choice task, consisting of a single text prompt and four images, each generated by the baselines and SynGen. Raters could also indicate that all images are "equally good" or "equally bad". We provided each prompt and its corresponding generations to three raters, and report the majority decision. In cases where there is no majority model winner, we count it toward "no majority winner".

We evaluate generated images in two main aspects: (1) *concept separation* (sometimes known as editability [17]) and (2) *visual appeal*. Concept separation refers to the ability of the model to distinctly depict different concepts or objects in the generated image. The effectiveness of concept separation is assessed by asking raters, "Which image best matches the given description?". To asses visual quality, raters were asked "Which image is more visually appealing?". To maintain fairness and reduce biases, the order of images was randomized in each task. Full rater instructions and further details are provided in Appendix G.1 of the supplemental materials.

We also experimented automatic evaluation, but find its quality subpar. For standardized evaluation purposes, it is detailed in Appendix G.2.

**Fine-grained evaluation.**   In addition to a multiple-choice task, we evaluate concept separation using the following key metrics: (1) Proper Binding, quantifying how well the model associates attributes with their corresponding objects; (2) Improper Binding, measuring the instances where attributes are incorrectly linked to unrelated objects; and (3) Entity Neglect, capturing the frequency with which the model omits entities specified in the prompt.

To this end, we randomly select 200 prompts each from the DVMP and ABC-6K datasets, while using all 177 prompts available in the A&E dataset. Human evaluators were asked to mark if instances have correct or incorrect attribute-object mapping. Importantly, incorrect mappings are counted on a per-attribute basis—multiple incorrect mappings of a single attribute are considered one violation. For example, in the prompt "the white dog chased the cat up the tree", if the modifier "white" is incorrectly mapped to both "cat" and "tree", it is counted as one instance of violation. Evaluators also identify the number of entities mentioned in the prompt that are subsequently depicted in the generated image.

Based on these counts, we define the metric of *Proper Binding* as the ratio of correctly mapped attributes to the total number of attributes. Similarly, *Improper Binding* is defined as the ratio of incorrectly mapped attributes to the total number of attributes, while Entity Neglect is the complement of the ratio of mentioned entities that are depicted in the generated image to the total number of entities in the prompt. Rater instructions are provided in Appendix G.1.

## 4   Results

### 4.1   Quantitative Results

Table 1 provides results of the comparative experiment. SynGen is consistently ranked first in all three datasets, and by a large margin, sometimes double the approval rate of the second ranked method, A&E. These results are observed for concept separation, which measures directly the semantic leak, and for visual appeal.

The high number of "no winner" cases reflects the large difficulty of some of the prompts, for which no method provides good enough generated images. Population results before majority aggregation are given in Appendix G.1 of the supplemental material. Comparisons with StableDiffusion are given in Fig. 19 of the supplemental.

Table 2 provides results of the individual experiment. We find that SynGen outperforms all models by a landslide in both proper and improper binding and is on par with state-of-the-art on entity neglect [11], despite not directly tackling this problem.

| Dataset | Model | Concept Separation | Visual Appeal |
|---|---|---|---|
| A&E | SynGen (ours) | **38.42** | **37.85** |
| | A&E | 18.08 | 18.65 |
| | Structured Diffusion | 04.52 | 04.52 |
| | Stable Diffusion | 01.69 | 02.26 |
| | No majority winner | 37.29 | 36.72 |
| DVMP (challenge set) | SynGen (ours) | **24.84** | **16.00** |
| | A&E | 13.33 | 12.17 |
| | Structured Diffusion | 04.33 | 07.83 |
| | Stable Diffusion | 03.83 | 07.17 |
| | No majority winner | 53.67 | 56.83 |
| ABC-6K | SynGen (ours) | **28.00** | **18.34** |
| | A&E | 11.17 | 10.00 |
| | Structured Diffusion | 05.83 | 06.33 |
| | Stable Diffusion | 04.83 | 07.83 |
| | No majority winner | 50.17 | 57.50 |

Table 1: Human evaluation of all methods on the three datasets. The table reports scores for concept separation (how well the image matches the prompt) and visual appeal. Values are the fraction of majority vote of three raters, normalized to sum to 100.

Table 2: Results of the fine-grained concept separation experiment. Proper Binding should be maximized to 100, while Improper Binding and Entity Neglect should be minimized to 0.

| Dataset | Model | Proper Binding ↑ | Improper Binding ↓ | Entity Neglect ↓ |
|---|---|---|---|---|
| A&E | SynGen (ours) | **94.76** | **23.81** | 02.82 |
| | A&E | 81.90 | 63.81 | **01.41** |
| | Structured Diffusion | 55.71 | 67.62 | 21.13 |
| | Stable Diffusion | 59.05 | 68.57 | 20.56 |
| DVMP (challenge set) | SynGen (ours) | **74.90** | **19.49** | 16.26 |
| | A&E | 52.47 | 31.64 | **10.77** |
| | Structured Diffusion | 48.73 | 30.57 | 28.46 |
| | Stable Diffusion | 47.80 | 30.44 | 26.22 |
| ABC-6K | SynGen (ours) | **63.68** | **14.37** | 34.41 |
| | A&E | 56.26 | 26.43 | **33.18** |
| | Structured Diffusion | 51.47 | 29.52 | 34.57 |
| | Stable Diffusion | 52.70 | 27.20 | 36.57 |

## 4.2 Qualitative Analysis

Figures 4–6 provide qualitative examples from the three datasets, comparing SynGen with the two strongest baselines.

The qualitative examples illustrate several failure modes of our baselines. First, ***semantic leak in prompt***, occurs when a modifier of an entity-noun "leaks" onto a different entity-noun in the prompt, as shown in Fig. 4, for the prompt "a pink clock and a brown chair", in columns 3 and 4. In this case, all baselines incorrectly apply pink hues to the chair, despite the prompt explicitly defining it as brown. A more nuanced variant of this issue is ***semantic leak out of prompt***, when a modifier is assigned to an entity-noun that is not mentioned in the prompt. For instance, the "spiky" attribute in "a spiky bowl and a green cat" leaks to a plant, which is not in the prompt, or the green coloration in the background of the images generated by the baselines, as seen in columns 5 and 6 in Fig. 5.

***Attribute neglect*** occurs when a modifier from the prompt is absent from the generated image. As exhibited in Fig. 4, for "a frog and a brown apple", both baselines do not include a brown color at all.

***Entity casting*** is another failure type where a modifier is treated as a standalone entity, a phenomenon commonly observed with noun modifiers. For example, the prompt "a wooden crown and a furry *baby*

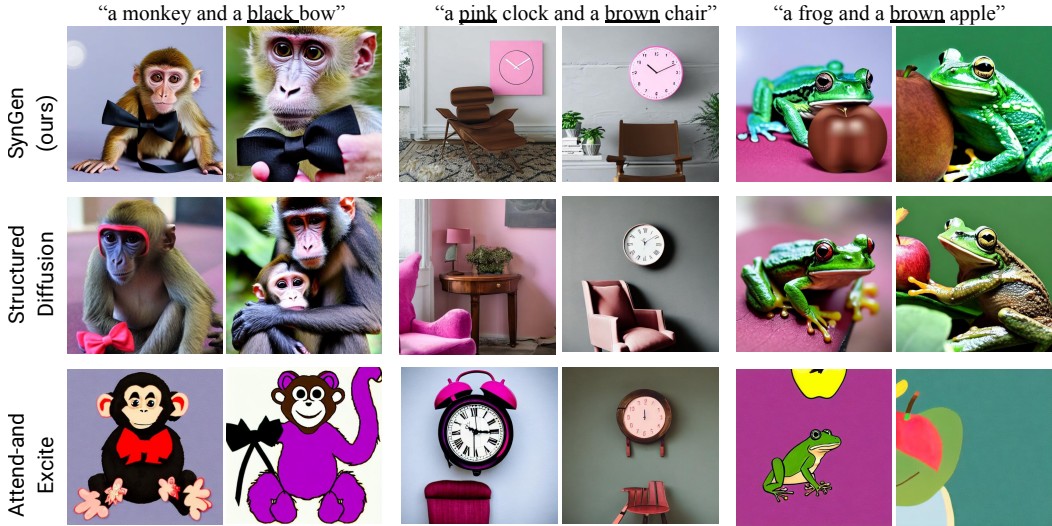

Figure 4: Qualitative comparison for prompts from the Attend-and-Excite dataset. For every prompt, the same three seeds are used for all methods.

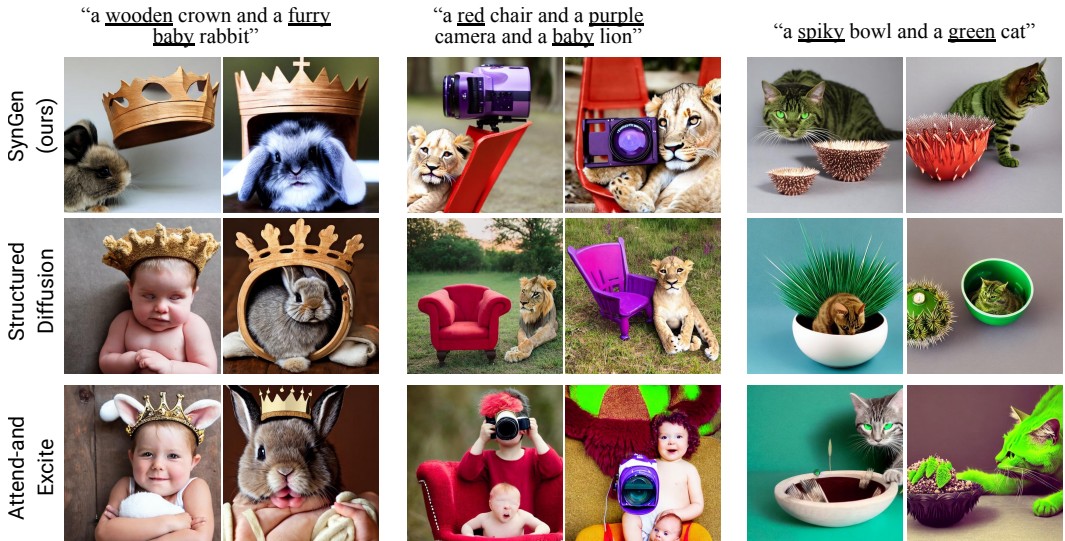

Figure 5: Qualitative comparison for prompts from the DVMP dataset. For every prompt, the same three seeds are used for all methods.

rabbit" (column 1 in Fig. 5) has all methods, apart from ours, generate human infants. Presumably, this occurs because "baby" is interpreted as a noun rather than as a modifier, leading other methods to treat it as a separate object due to the lack of syntactic context. Conversely, SynGen correctly interprets "baby" as a modifier and accurately binds it to the rabbit. Similarly, in the prompt "a white fire hydrant sitting in a field next to a red building" (column 6 in Fig. 6), "fire" is wrongly interpreted as an entity-noun, which leads to the unwarranted inclusion of a fire in the scene.

All methods, barring SynGen, grapple with ***entity entanglement*** [18, 19, 20, 21, 22], where some objects tend to strongly associate with their most common attribute (e.g., tomatoes are typically red). This is evident in columns 3 and 4 in Fig. 6, where other methods fail to visually associate the blue attribute with the dog in "a blue and white dog sleeps in front of a black door". Instead, they resort to typical attributes of the objects, generating a black and white dog.

Further qualitative analysis is provided in Appendix D.1.

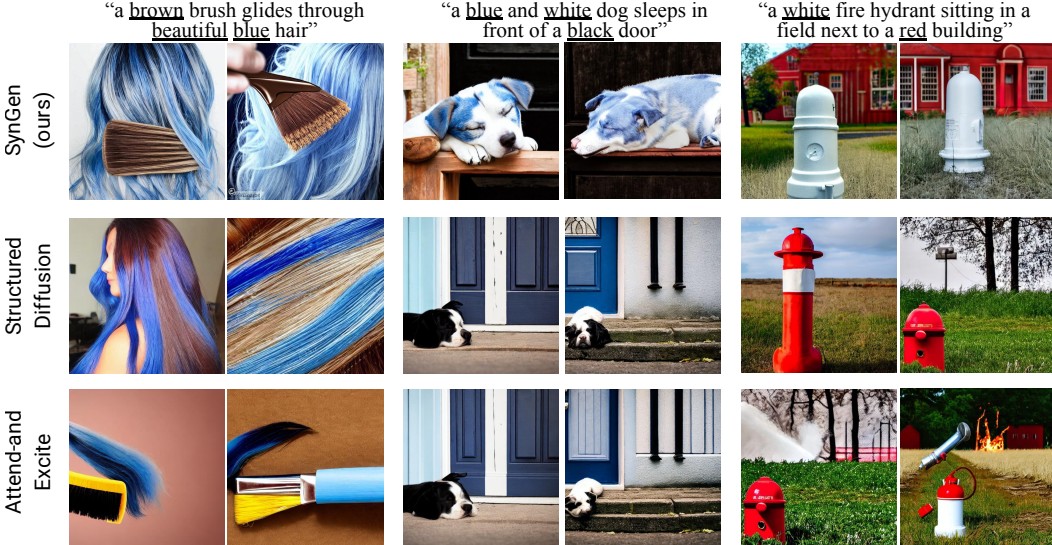

Figure 6: Qualitative examples for ABC-6K prompts. For every prompt, all methods use the same three seeds.

## 4.3   Ablation study

**The importance of using both positive and negative losses.**    We evaluated the relative importance of the two terms in our loss Eq. (3). The positive term $\mathcal{L}_{pos}$, which encourages alignment of the attention map of an object and its modifiers, and the negative loss term, $\mathcal{L}_{neg}$, which discourages alignment with other modifiers and objects. We sampled 100 prompts from the DVMP dataset and generated images with and without each of the two loss terms. See example in Fig. 7. Then, raters were asked to select the best of four variants. Table 3 shows that raters preferred the variant that combined both the positive and the negative terms. More examples are given in the supplemental Appendix B.

| Loss | Concept Separation | Visual Appeal |
|---|---|---|
| Both losses $\mathcal{L}_{pos}+\mathcal{L}_{neg}$ | **27** | 22 |
| Positive only $\mathcal{L}_{pos}$ | 0 | 11 |
| Negative only $\mathcal{L}_{neg}$ | 3 | **35** |
| Stable Diffusion | 4 | 28 |
| No majority winner | 66 | 4 |

Table 3: **Ablation of loss components**. Values are percent preferred by human raters.

| Both Losses | Only Positive Loss | Only Negative Loss | No Intervention |
|---|---|---|---|

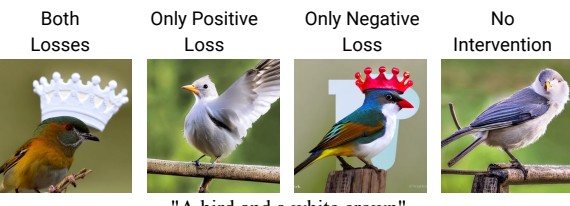

"A bird and a white crown"

Figure 7: **Ablation of loss components.** Removing $\mathcal{L}_{neg}$ results in semantic leakage (the bird is white) and entity neglect (there is no crown). Removing $\mathcal{L}_{pos}$ also leads to semantic leakage (generating a bird and background with white parts), and failed attribution binding (generating a crown that is not white).

## 5   Related Work

**Semantic leakage**. [2] pointed out cases of semantic leakage in diffusion models, where properties of one entity-noun influence the depiction of another. [23] attributed this issue to a lack of understanding of syntax, specifically noting failures when processing texts requiring subtle syntactic binding comprehension. [6] identified semantic leakage issues in DALL-E, where properties of one entity-noun influence how other entity nouns are depicted. In this work, we pinpoint semantic leakage as a consequence of improper mapping between syntactic and visual binding.

**Attention-based interventions.** [15] demonstrated that the cross-attention mechanism determines the spatial layout of entities in generated images. This result suggested that cross-attention is causally involved in the aforementioned issues. A&E [11] addresses the problem of entity omission, where certain entities mentioned in the prompt do not appear in the generated image. They propose a loss function that encourages each noun token in the image to significantly attend to a corresponding image patch, thereby preventing its omission. Our approach is similar to [11] in that it updates the latent representation through a loss function over attention maps, during image generation.

Syntax-based generation was also explored in [13], proposing the Structured Diffusion method. It aims to address the problem of missing entities and semantic leakage of attributes. This is achieved by parsing the prompt, extracting phrases corresponding to nouns and modifiers, and encoding them separately. They also intervene in the attention patterns, ensuring that each individual phrase influences the attention patterns. Our experiments show that it is better to implicitly influence the attention patterns through our loss which we dynamically optimize. In contrast, their intervention remains fixed.

Concurrent to this work, [24] proposed an alternative approach to combine syntactic control and attention-based optimization. They extract nouns from prompts and train a layout predictor to identify the corresponding pixels for each noun. Then, they optimize the latents by encouraging the pixels corresponding to the objects to attend to CLIP representations of phrases containing those objects. While similar in spirit, the current paper demonstrates intervention in the generation process solely based on syntax, without explicitly learning the correspondence between image entities and tokens.

## 6 Limitations

Like previous methods, the performance of SynGen degrades with the number of attributes to be depicted (see supplemental Fig. 12). However, its decline is remarkably less pronounced compared to other methods. This decay in performance can be attributed to two primary factors: (1) an image begins to lose its visual appeal when the negative loss term becomes excessively large; (2) an overly cluttered image poses challenges in crafting a cohesive "narrative" for all the concepts. We expect that some of these issues can be addressed with more hyper-parameter tuning.

Naturally, the effectiveness of our method is intrinsically tied to the quality of the parser. When the parser fails to extract the stipulated syntactic relations, our method essentially operates akin to SD.

Finally, SynGen takes longer to generate images with modifiers in the prompt than SD and slightly slower than than A&E (see Appendix A).

## 7 Conclusions

In this work, we target the improper binding problem, a common failure mode of text-conditioned diffusion models, where objects and their attributes incorrectly correspond to the entity-nouns and their modifiers in the prompt. To address it, we propose SynGen, an inference-time intervention method, with a loss function that encourages syntax-related modifiers and entity-nouns to have overlapping cross-attention maps, and discourages an overlap from cross-attention maps of other words in the prompt. We challenge our method with three datasets, including DVMP – a new dataset that is specially-designed to draw out hard cases of improper-binding problem. Our method demonstrates improvement of over 100% across all three datasets over the previous state-of-the-art. Finally, our work highlights the importance of linguistic structure during denoising for attaining faithful text-to-image generation, suggesting promising avenues for future research.

## Acknowledgements

This study was funded by a grant to GC from the Israel Science Foundation (ISF 737/2018) and an equipment grant to GC and Bar-Ilan University from the Israel Science Foundation (ISF 2332/18). This project has also received funding from the European Research Council (ERC) under the European Union's Horizon 2020 research and innovation programme, grant agreement No. 802774 (iEXTRACT). Shauli Ravfogel is grateful to be supported by the Bloomberg Data Science PhD Fellowship.

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

# Supplementary Material

## A    Implementation Details

**Computing resources.**    Experiments were run on an NVIDIA DGX Station with four v100-SXM2-32GB GPUs. The overall duration of all experiments in the paper was about two weeks.

**Efficiency.**    SynGen takes ~144% longer than SD and ~10.9% longer than A&E to generate images with modifiers in the prompt. To arrive to these numbers, we randomly sampled a set of 50 images from the A&E dataset, the DVMP set, and ABC-6K and timed the generations for each method. On average, SD needs 4 seconds to generate an image, A&E 8.8 seconds, and SynGen 9.76 seconds.

**Hyperparameters.**    The hyperparameters we used consist of 50 diffusion steps, a guidance scale of 7.5, a scale-factor of 20, and 25 latent update steps. The choices of scale factor and latent update steps are described in Appendix B.2 and Appendix B.3 respectively.

**Parser.**    Throughout this project, we use the spacy parser with the out-of-the-box `en_core_web_trf` model.

**Attending word pieces.**    When a relatively uncommon word is encountered by the tokenizer of the text encoder, it is split to sub-words (i.e., word pieces). In the context of our loss function, when an entity (or modifier) is split into word pieces, we compute our distance function (the Symmetric-KL) for each word piece. Then, only the word piece that maximizes the distance is added to the loss.

**Cross-attention maps.**    We describe more formally the cross-attention maps on which we intervene. Let $N$ be the number of tokens in the prompt, and let $D^2$ be the dimensionality of the latent feature map in an intermediate denoising step. The denoising network defines a cross-attention map $A^{patches \to tokens} \in \mathbb{R}^{D^2 \times N}$ between each of $D^2$ patches in the latent feature map and each token. Intuitively, the attention maps designates which tokens are relevant for generating each patch.

Our goal it to derive an attention distribution $A^{tokens \to patches} \in R^{N \times D^2}$ such that its $i$-th row $A_i^{tokens \to patches}$ contains the attention distribution of token $i$ over patches. For this goal, we define a score matrix $S$ to be the transpose of $A^{patches \to tokens}$, i.e,. a matrix whose $i^{th}$ row contains the attention scores from each patch to token $i$. Since $S$ is not normalized, we divide each row by its sum to get a distribution over patches. Unless stated otherwise, across the paper, we refer to $A^{tokens \to patches} \in R^{N \times D^2}$ when mentioning the "cross-attention maps" $A$ and its $i^{th}$ row $A_i$ corresponding to the attention map from the $i^{th}$ token.

## B    Additional Ablation Experiments

### B.1    Further Investigation of the Positive and Negative Loss Terms

In Section 4.3, we investigate the importance of the positive and negative loss function terms using a human rater. Here, we accompany the rating with a qualitative analysis, to examine the effect of each term. To this end, we generate images for 15 randomly selected prompts, five from each dataset. Fig. 8 depicts a sample of the generated prompts.

We find that proper binding necessitates both the positive and negative terms: excluding the negative term from the loss function results in two noteworthy observations. First, the number of missing objects increase, evident by the missing crown, cat, metal chair, and tomato, in columns 1, 2, 4, and 5 in Fig. 8. One consequence of missing objects is the apparent improper binding, indicated by the red backpack and black shirt in columns 1 and 3.

On the other hand, excluding the positive term results in fuzzier separation between objects. For instance, the cat is not completely formed, and is "merged" with the pillow; and while it appears that there is some green residue on the dog, it is not colored green. Moreover, the grass is green, which indicates a semantic leakage.

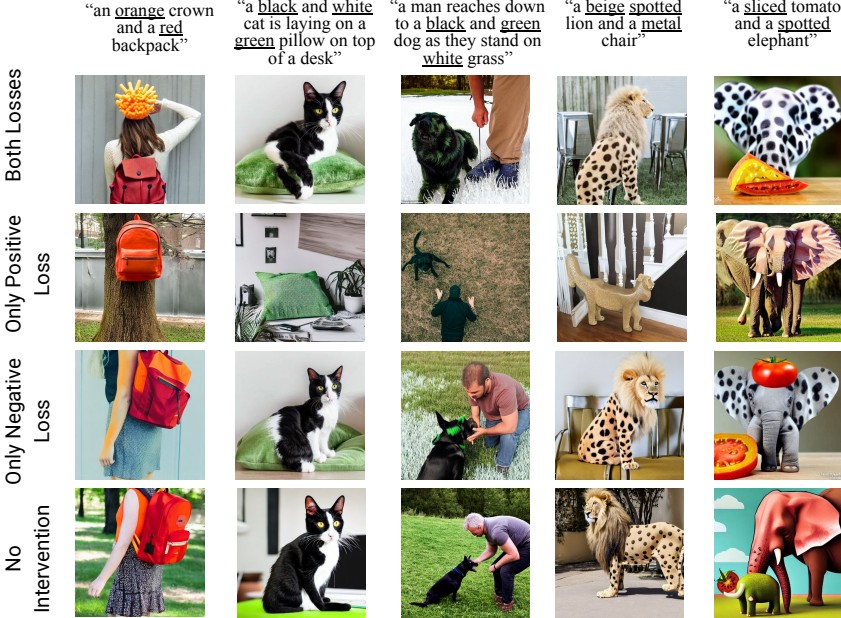

Figure 8: We examine the effect of employing only one of the two losses instead of both. All images were generated using the same random seed.

Putting these insights together, we observe that to some extent, the effect the loss terms is complementary. In addition to the increase of objects and proper binding, the images are more coherent (less cases of objects mixed into each other, such as the cat in the only-negative loss or the elephant in the only-positive loss).

## B.2 Number of Timesteps for Intervention

Recall that our method intervenes in latent denoising generation. In this appendix, we study the effect of the hyperparameters determining the number of steps in which we intervene.

To identify an ideal number of timesteps to intervene, we experiment with 100 randomly selected prompts from the DVMP dataset, a fixed random seed, and a number of update steps from 5 to 50, in increments of 5. Examples of this experiment are shown in Fig. 9.

We observe that when intervening in a small number of timesteps, our method failed to adequately mitigate semantic leakage or that images are not completely formed. For instance, the apple in column 1 in the 15-steps example is cartoon-ish, while the dog is not. Conversely, intervening for the full 50 timesteps resulted in an increase rate of blurred images (potentially due to the significant modification of the latent, which shifts it away from the learned distribution). We conclude that the optimal number of timesteps for intervention is 25, as this allows for effective mitigation of improper binding, while still generating visually appealing images.

## B.3 Setting the Scale Factor

The scale factor affects the update step size. Recall the update step stated in Section 2 $z't = z_t - \alpha \cdot \nabla z_t \mathcal{L}$. Here, $\alpha$ is the scale-factor.

To determine a good selection for the scale-factor, we generate 100 randomly sampled prompts from the DVMP dataset, with a scale-factor value from 1 to 40, in increments of 10.

As can be seen in Fig. 10, we observe that merely updating the latent using a scale-factor of 1 yields relatively good results in terms of improper binding, which confirms the utility of our loss function. However, such a low scale-factor also consistently leads to missing objects.

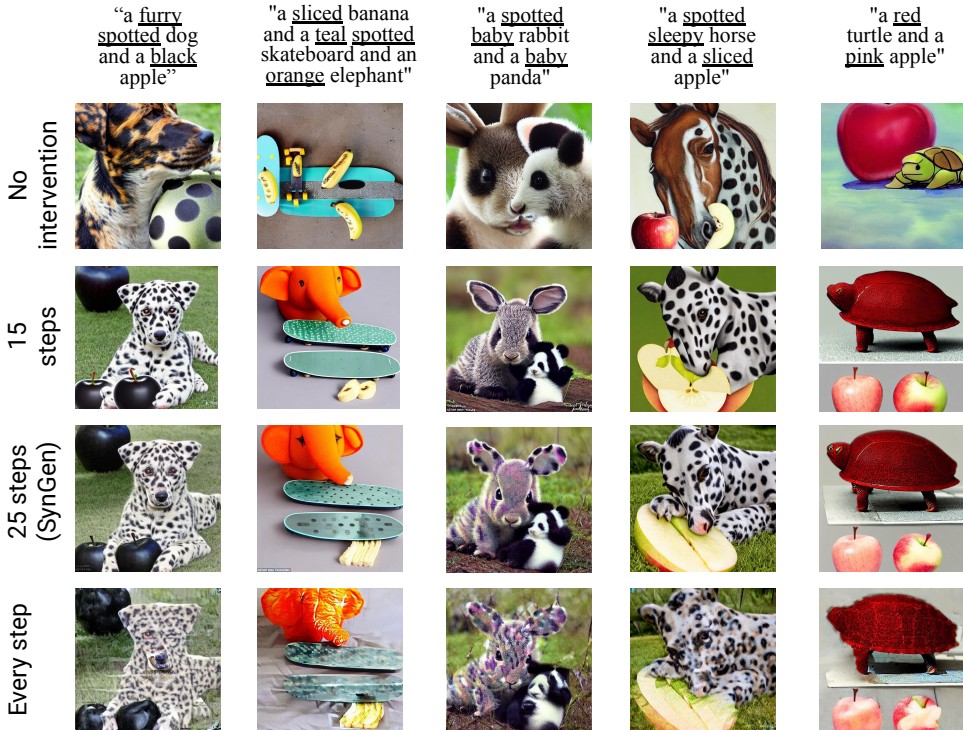

Figure 9: We experiment with varying number of diffusion steps and examine the effect of changing the number of diffusion steps for which we intervene with the cross attention maps. All images were generated using the same random seed.

Interestingly, for greater scale-factor values, the generations become alike in their overall look, but are nonetheless very different. As an example, for both values, 10 and 30, the sliced banana is missing from the image in column 2, but the 30-value does result in a spotted teal skateboard. In column 3, values below 20 lead to images that contain two pandas (none of which are spotted), which indicates the proper binding process, and that the latent was not updated enough. On the other hand, a value greater than 20 leads to an image of a striped rabbit, instead of a spotted rabbit.

One interesting conclusion from this experiment is that the greater the scale-factor, the stronger the concept separation. However, this is only true to a point. For a great enough value, generations become too blurred or simply lose their visual appeal.

## C  Additional Quantitative Analyses

To study the efficacy of SynGen relative to the baselines in improper binding setting, we analyze the results under three perspectives. (1) as a function of repeating entities and modifiers; (2) as a function of the number of modifiers; and (3) degree of entanglement. Samples of generations are shown in Fig. 14.

**Number of repeating modifiers and entities.**    In this analysis, we examine the performance of all methods for prompts containing recurring modifiers (e.g., "a *sliced* strawberry and a *sliced* tomato) or entities (e.g., "a sliced *tomato* and a skewered *tomato*"). Aggregated results are illustrated in Fig. 11. Our observations reveal a decrease in performance across all methods when modifiers are repeated. However, the relative success between SynGen and the baselines in performance remains the same. Moreover, there is no substantial decline in results when entities are repeated.

**Number of modifiers in prompt.**    We hypothesize that since our method is specifically designed to tackle improper binding, it handles prompts containing many modifiers with more success. This is

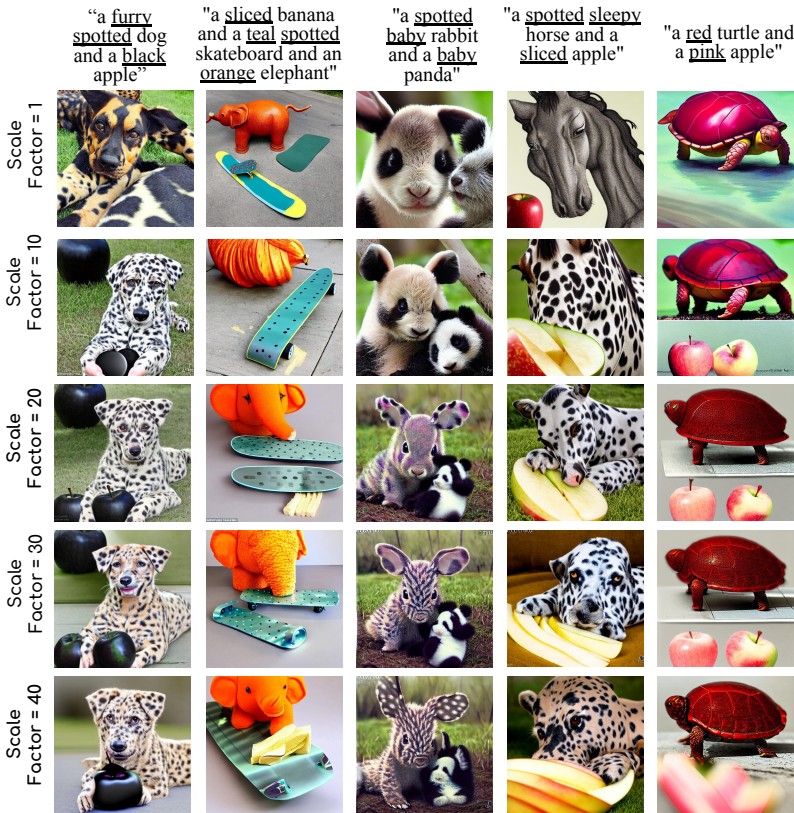

Figure 10: Qualitative comparison between scale factor values for SynGen. For every prompt, the same seeds are applied. We anecdotally show our scale-factor value (we use the value 20) provides superior results.

confirmed in Fig. 12, which shows the gap between SynGen and the baselines widens as the number of modifiers in a prompt increases.

**Entangled entities.** As we describe in Section 4.2, entangled entities are strongly associated with their most common attribute. For instance, a tomato is typically red, and thus, it is common for images to depict *red* tomatoes.

We categorize the prompts into three groups: (1) entangled prompts, which contain entangled entities with a modifier that overrides a common modifier (e.g., a *purple* strawberry); (2) common entangled prompts, which contain entangled entities with their common modifiers; and (3) neutral prompts, which do not contain entangled entities at all. Performance as a function of these groups is demonstrated in Fig. 13.

# D   Additional Qualitative Results

## D.1   Qualitative analysis by number of modifiers

In Fig. 15, examples from the DVMP challenge set include 2 to 6 modifiers.

While errors of all types are prevalent regardless of the number of modifiers, their frequency tends to rise as more modifiers are added.

As for SynGen, although it does not display semantic leakages at an increased rate compared to the baselines (as quantitatively demonstrated in Fig. 12), it does show a tendency to generate more than the specified number of entities as the modifier count increases. This behavior is observable in rows 8 and 10 for SynGen, and in rows 7 through 10 for the baselines.

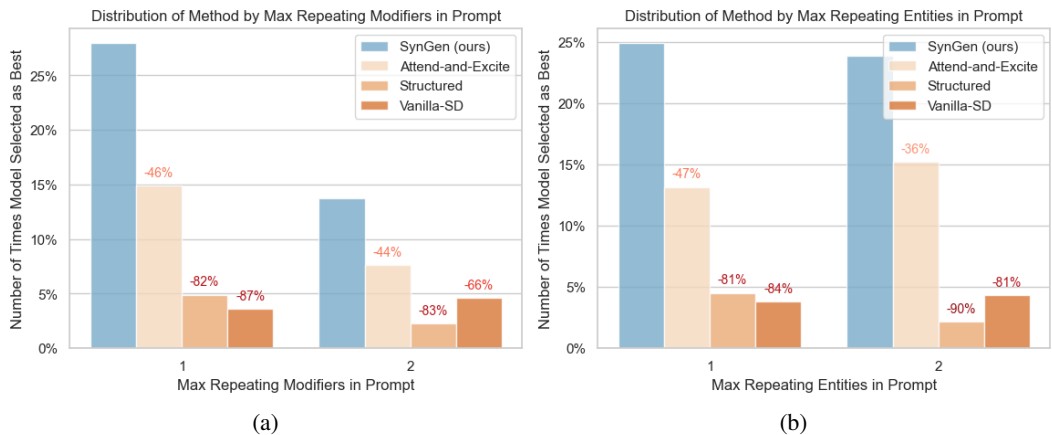

Figure 11: The performance of SynGen and the baselines in concept separation on prompts containing (a) repeating modifiers; and (b) repeating entities in the DVMP dataset.

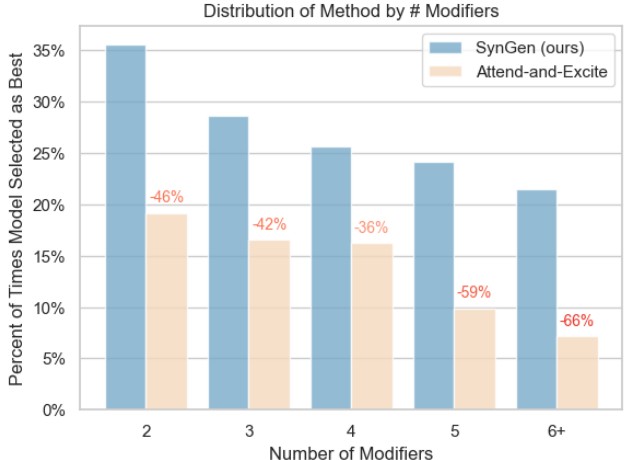

Figure 12: Concept Separation as a function of number of modifiers in a prompt in the DVMP dataset, introduced in Section 3.2. Only the top-competing method (Attend-and-Excite) is plotted for readability.

## D.2 Comparison to Spatial-Temporal Diffusion

As described in Section 5, concurrent to this work, [24] developed a method to optimize the latents. While they primarily attend spatial and temporal relations, they too report on improper binding, namely, attribute mismatch. Thus, we extend the tables from Section 4, to include Spatial-Temporal Diffusion, see Fig. 16, Fig. 17, Fig. 18.

Based on these 18 images, we observe that Spatial-Temporal Diffusion consistently misses at least one entity from the prompt. As an example, see Fig. 16. The images in columns 1 and 2 miss a crown (but include "wooden" objects), and columns 3 and 4 miss a lion and exhibit semantic leakage.

In other cases, we note many cases of semantic leakage in and out of the prompt. For instance, in Fig. 18, in column 2 the clock is brown and the wall is pink, and in column 3, the chair is pink.

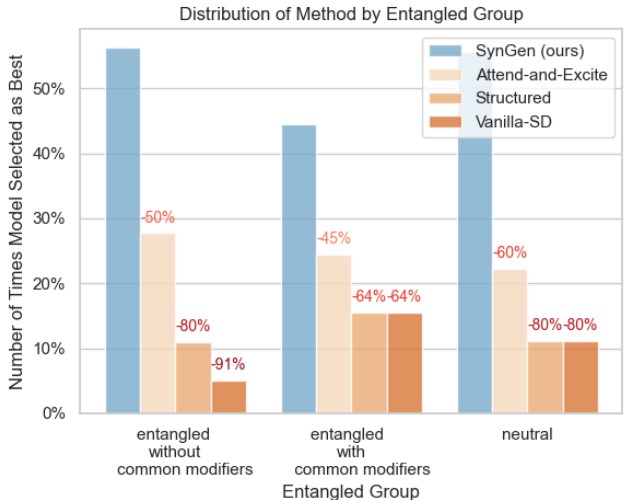

Figure 13: The performance of SynGen and the baselines in concept separation when grouping the prompts with respect to entangled modifiers in the DVMP dataset.

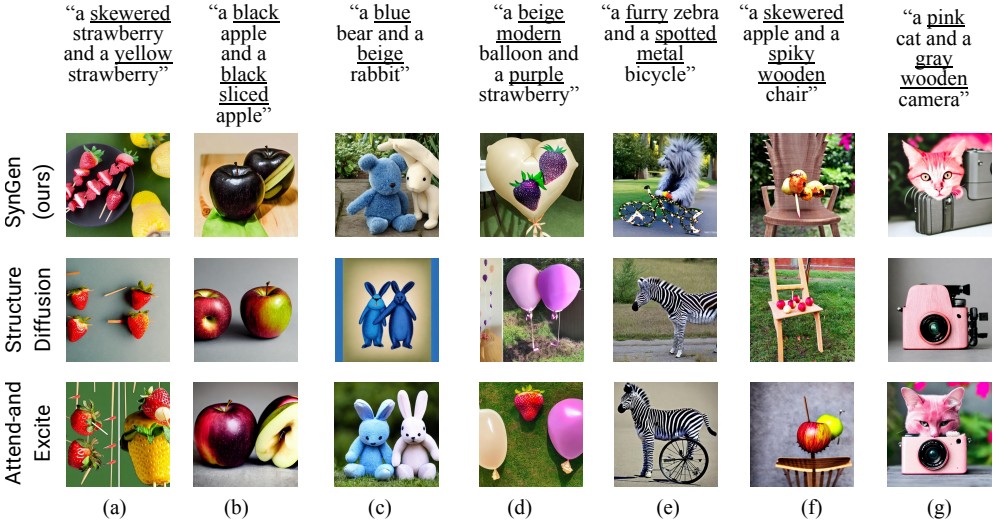

Figure 14: Samples from the analyses in Appendix C. (a) a case of recurring entity (strawberry); (b) a recurring modifier (black) and entity (apple); (c) and (d) contain entangled entities (a blue bear and a purple strawberry); (e), (f), (g) are examples of prompts with more than two modifiers.

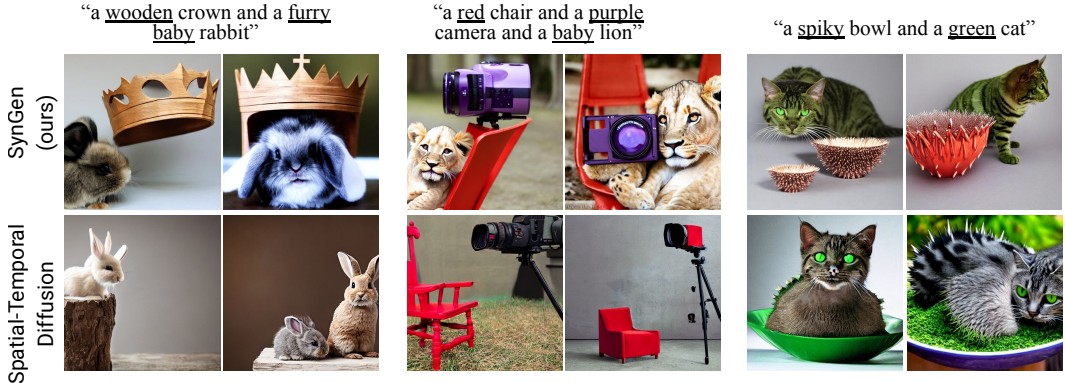

Figure 16: Extended qualitative comparison for prompts from the DMVP dataset. SynGen and Spatial-Temporal Diffusion [24].

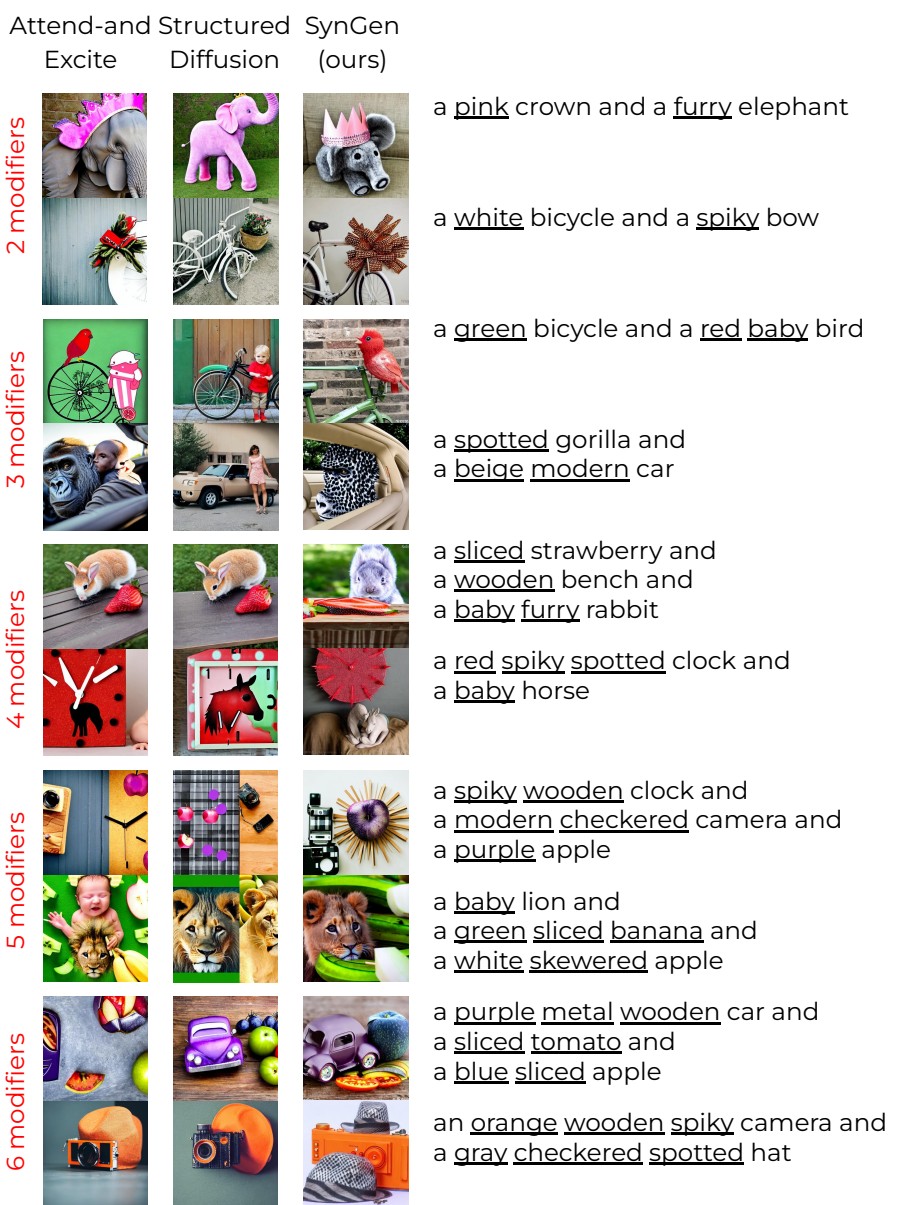

Figure 15: Extended qualitative comparison for prompts from the DVMP challenge set.

## D.3 Stable Diffusion and Structured Diffusion Comparison

A comparison between Stable Diffusion and Structured Diffusion is depicted in Fig. 19. The findings from the study by [11] suggest that the generated images from Structured Diffusion are often similar to those generated by Stable Diffusion, with limited improvements in addressing semantic flaws and enhancing image quality. This is further supported by the comparable results presented in our findings Table 1. Therefore, while we include all baselines in our evaluations, our qualitative analysis only showcases images produced by the slightly superior Structured Diffusion.

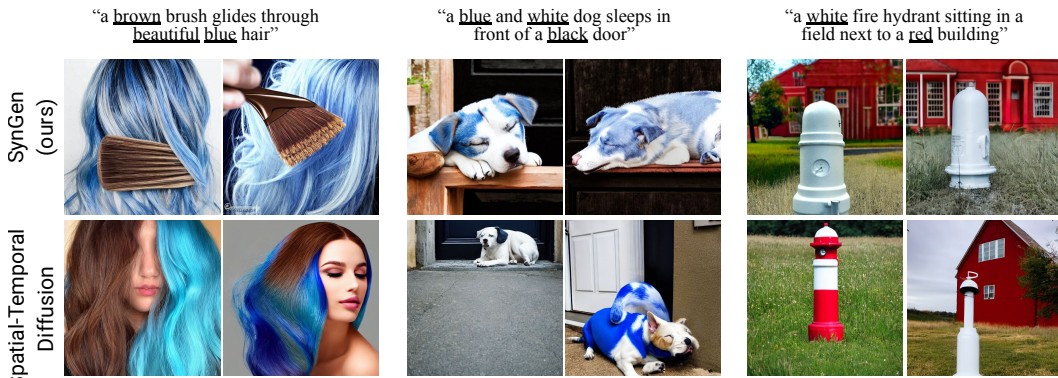

Figure 17: Extended qualitative comparison for prompts from the ABC6K dataset. SynGen and Spatial-Temporal Diffusion [24].

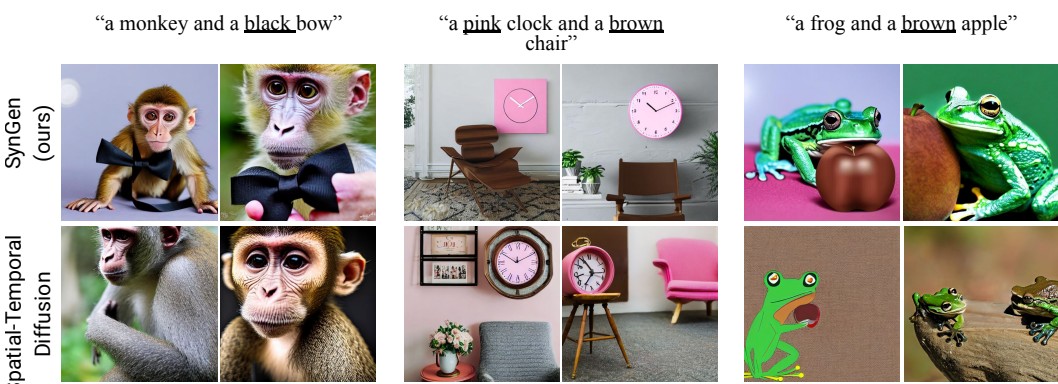

Figure 18: Extended qualitative comparison for prompts from the Attend-and-Excite dataset. SynGen and Spatial-Temporal Diffusion [24].

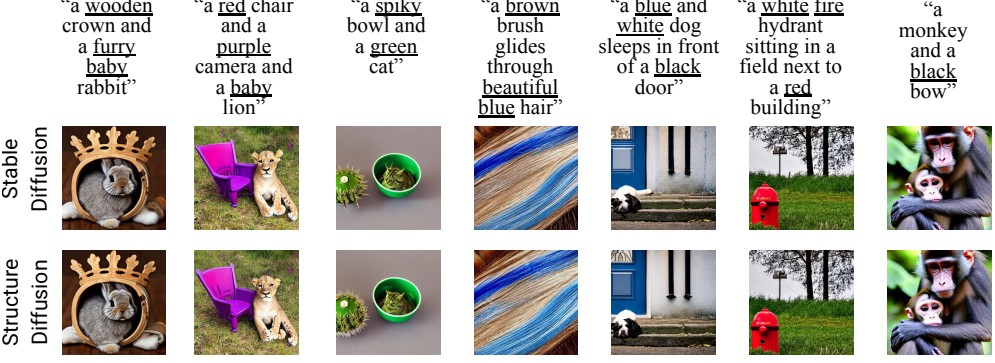

Figure 19: Side-by-side generations of StableDiffusion and StructureDiffusion.

# E SynGen Failures

We observe three recurring types of failure SynGen displays Fig. 20. First, when there are many modifiers and entities in the prompt, despite the results in Fig. 12, we note that sometimes the negative loss component becomes exceedingly large, and thus, pushes the latent out of the distribution the decoder was trained on. Consequently, images become blurred, or contain concepts which are successfully separated, but are incoherent. This is likely because our method over-fixates on incorporating all elements described in the prompt.

Second, while SynGen typically successfully addresses the possible error cases described in Section 4.2, at times it can neglect generating all objects, unify separate entities, or neglect generating attributes. We conjecture that it is because the cross-attention maps of the modifier and its corresponding entity do not overlap enough. We note it usually occurs when there are many modifiers that refer to the same entity.

Finally, as common with many diffusion models, we report a recurring issue with faithfulness to the number of units specified in the prompt, for a certain entity. For instance, upon receiving prompts containing "a strawberry", SynGen generates images with multiple strawberries, instead of just one. One explanation to this problem is that the representation of a certain entity begins "scattered", and is never quite formed into a single cluster. Interestingly, the opposite problem, where multiple units are "merged" into one, occurs far less in the generations of SynGen. Possibly, because of the inherent objective function of our loss, which "pushes away" foreign concepts from one another.

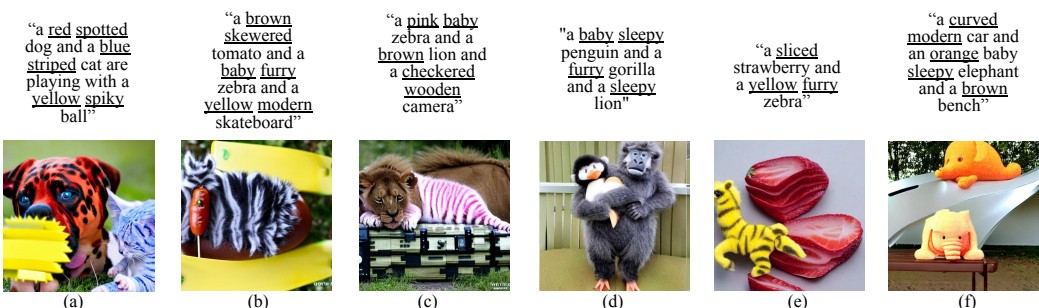

Figure 20: Frequent failure modes in SynGen. (a) depicts a case of blurred image, (b) incoherent image which maintains concept separation. Both are a result of excessive updates to the latent, resulting from a large negative loss term. In example (c), the zebra and lion are merged into a single entity and (d) omits the sleepy lion. We conjecture (c) and (d) are a result of too little updates. (e) and (f) exhibit the well-known issue of flawed mapping between the number of units an entity is mentioned in the prompt to the generated image.

## F  The Diverse Multiple Modifiers Prompts (DVMP) dataset

In Section 3.2 we describe DVMP, a new dataset containing rich and challenging combinations, for the purpose of evaluating improper binding.

In total, DVMP has 18 types of objects, 16 types of animals, and 4 types of fruit. There are four animal modifiers, 7 object modifiers, two fruit modifiers, and 13 colors. A comprehensive account of the entities and their possible modifiers is shown in Table 4.

## G  Extended Evaluation

### G.1  Additional Details on Human Evaluation Experiments

In the manual evaluation procedure detailed in Section 3.3 the evaluator is tasked with comparing various image generations and selecting the optimal image based on multiple criteria. The guidelines and examples given to the evaluators are presented in Fig. 21 and Fig. 22. Fig. 23 provides a screenshot of the rating interface. The full results of the human evaluation are given in Table 5

**Rater Compensation.**  Raters were selected based on their performance history, requiring a minimum of 5,000 approved HITs with an approval rate exceeding 98%. They were required to pass a qualification exam with a perfect score before given access to the task. The hourly compensation was $10, ensuring fair renumeration for their contributions.

Table 4: List of entities and their modifiers in the DVMP dataset. Colors are not restricted to categories.

| Category | Entities | Modifiers |
|---|---|---|
| General | backpack, crown, suitcase, chair, balloon, bow, car, bowl, bench, clock, camera, umbrella, guitar, shoe, hat, surfboard, skateboard, bicycle | modern, spotted, wooden, metal, curved, spiky, checkered |
| Fruit | apple, tomato, banana, strawberry | sliced, skewered |
| Animals | cat, dog, bird, bear, lion, horse, elephant, monkey, frog, turtle, rabbit, mouse, panda, zebra, gorilla, penguin | furry, baby, spotted, sleepy |

| Color Modifiers |
|---|
| red, orange, yellow, green, blue, purple, pink, brown, gray, black, white, beige, teal |

**Instructions**

- Please read the following instructions carefully:
- In this task, you will be given a description and two images. Your job is to evaluate the images based on two criteria:
- **1. Concept Separation:** How well does the image match the given description?
- **2. Visual Appeal:** Which image looks overall better or more natural?
- **Concept Separation:** For each image, ask yourself:
  - Do you see all objects from the description?
  - Are all objects' details correct?
  - Are there any details on objects that should not be there?
  Choose the image that best matches the description. If they are equally good, choose 'equally good' and if they are equally bad, choose 'equally bad'.
- **Visual Appeal:** After evaluating Concept Separation, decide which image looks better or natural to you.
- Please choose the images based on Concept Separation first, and then consider their Visual Appeal.

Figure 21: The instructions that were given to the raters.

Table 5: The population vote of three raters was normalized to sum to 100 and the standard error mean was added. The table reports the scores for concept separation (how well the image matches the prompt) and visual appeal for different models on each dataset.

| Dataset | Model | Concept Separation | Visual Appeal |
|---|---|---|---|
| A&E | SynGen (ours) | **38.80** $\pm$ 0.48 | **40.11** $\pm$ 0.49 |
| | A&E | 22.60 $\pm$ 0.41 | 21.47 $\pm$ 0.41 |
| | Structured Diffusion | 09.98 $\pm$ 0.29 | 11.87 $\pm$ 0.32 |
| | Stable Diffusion | 08.85 $\pm$ 0.28 | 09.79 $\pm$ 0.29 |
| | No majority winner | 19.77 $\pm$ 0.39 | 16.76 $\pm$ 0.37 |
| DVMP (challenge set) | SynGen (ours) | **29.22** $\pm$ 0.45 | **23.55** $\pm$ 0.42 |
| | A&E | 19.83 $\pm$ 0.39 | 19.00 $\pm$ 0.39 |
| | Structured Diffusion | 09.00 $\pm$ 0.28 | 15.56 $\pm$ 0.36 |
| | Stable Diffusion | 09.89 $\pm$ 0.29 | 15.56 $\pm$ 0.36 |
| | No majority winner | 32.06 $\pm$ 0.46 | 26.33 $\pm$ 0.44 |
| ABC-6K | SynGen (ours) | **33.00** $\pm$ 0.47 | **25.72** $\pm$ 0.43 |
| | A&E | 17.84 $\pm$ 0.38 | 17.28 $\pm$ 0.37 |
| | Structured Diffusion | 13.44 $\pm$ 0.34 | 14.50 $\pm$ 0.35 |
| | Stable Diffusion | 11.72 $\pm$ 0.32 | 14.50 $\pm$ 0.35 |
| | No majority winner | 24.00 $\pm$ 0.42 | 28.00 $\pm$ 0.44 |

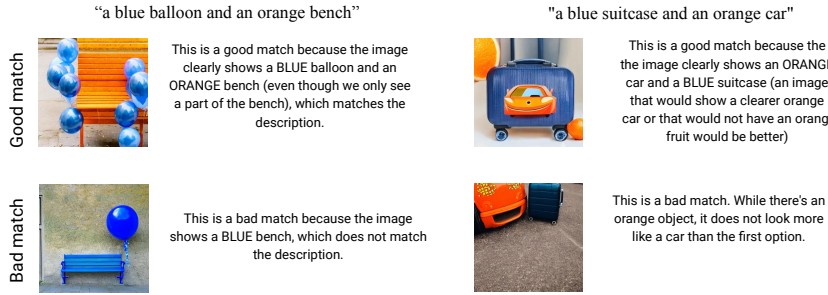

Figure 22: Examples given to raters in their instructions. Each example consists of a prompt and two images: A good match (top) and a bad match (bottom) for the concept separation criterion. These examples were accompanied by text explaining why the images are considered a good (or bad) match to the prompt.

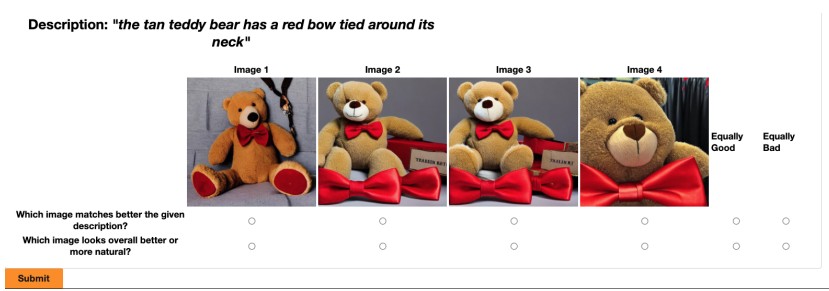

Figure 23: A screenshot of the AMT task. The order of images was randomized per HIT. "equally good" and "equally bad" were merged during post-processing into "no winner", to simplify presentation of results.

## G.2 Phrases-to-Image Similarity

A common approach to automatically assess text-based image generation is by computing the cosine similarity between an image and prompt, using a vision-language model like CLIP [9]. However, the very challenge we tackle here is rooted in CLIP's failure in establishing correct mapping between syntactic bindings and visual bindings, functioning like a bag-of-words model [10]. As an example, suppose CLIP is prompted with "a blue room with a yellow window". If we present CLIP with an image of a yellow room with a blue window, it may yield a similar score to an image that accurately depicts a blue room with a yellow window.

In an attempt to address this flaw, we segment prompts to phrases containing entity-nouns and their corresponding modifiers (e.g., "a blue room" and "a yellow window"), and compute the similarity between these segmented phrases and the image. We then aggregate the result to a single score by computing the mean. With this approach, we expect CLIP to properly associate the modifiers (e.g., "blue" and "yellow") with the correct entity-noun (i.e., "room" and "window") as there is only one entity-noun in each segment. Unfortunately, this metric achieves relatively low agreement with the majority selection of human evaluation, only 43.5% of the time, where 25% is random selection. Despite the low agreement, we note the overall trend of selections of this automatic metric is very similar to the human majority selection. Table 6 shows the results of our automatic evaluation.

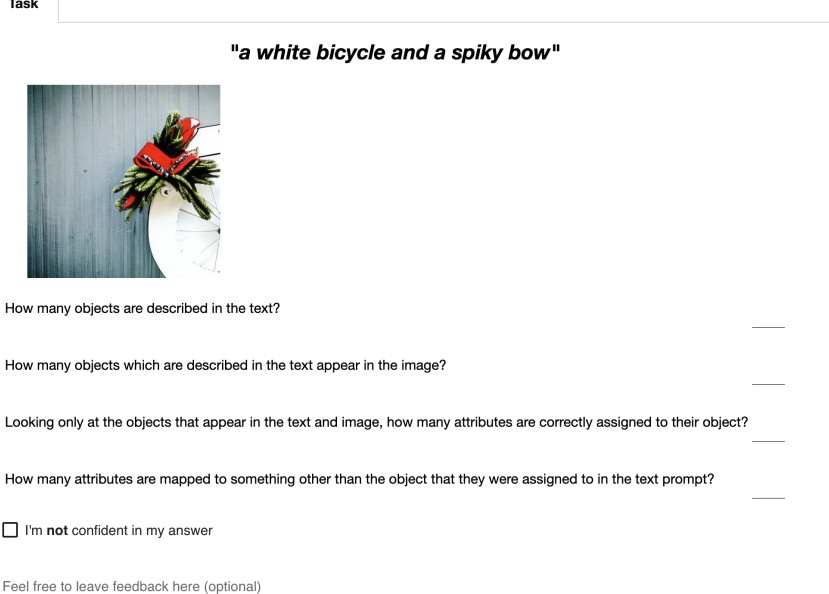

**"*a white bicycle and a spiky bow*"**

How many objects are described in the text?

How many objects which are described in the text appear in the image?

Looking only at the objects that appear in the text and image, how many attributes are correctly assigned to their object?

How many attributes are mapped to something other than the object that they were assigned to in the text prompt?

☐ I'm **not** confident in my answer

Feel free to leave feedback here (optional)

Figure 24: A screenshot of the fine-grained AMT task.

Table 6: Automatic evaluation of all methods on the three datasets. The table reports scores for concept separation (how well the image matches the prompt) and visual appeal. Values are the fraction of majority vote of three raters, normalized to sum to 100.

| Method | DVMP (ours) | ABC-6K | A&E |
|---|---|---|---|
| SynGen (ours) | **47.33** | **41.33** | **44.63** |
| A&E | 27.66 | 24.33 | 27.11 |
| Structured Diffusion | 12.84 | 17.84 | 11.87 |
| Stable Diffusion | 12.17 | 16.50 | 16.39 |

