# OpenReview forum: "Linguistic Binding in Diffusion Models: Enhancing Attribute Correspondence through Attention Map Alignment"
_NeurIPS.cc/2023/Conference — NeurIPS 2023 oral_

### Official Review · Reviewer_LWw2 · 2023-07-01

**Soundness:** 3 good
**Presentation:** 2 fair
**Contribution:** 3 good
**Rating:** 7
**Confidence:** 3

**Summary:**

This paper presents an approach called “SynGen” to improve attribute binding between nouns and their modifiers for text-to-image latent diffusion models, specifically Stable Diffusion. They propose a two-part loss on attention maps that (1) encourages attention maps of nouns and their modifiers to be similar, and (2) encourages attention maps between the noun/modifier to be different from those of other tokens in the prompt.

Datasets and baselines: They run experiments on three datasets: the attribute binding contrast set (ABC-6K) from the Structured Diffusion (baseline 1) paper, data from the Attend-and-Excite (baseline 2), and a newly proposed challenge set called Diverse Visual Modifier Prompts.

Evaluation:
(1) Human evals for concept separation and visual appeal. They find that SynGen outperforms baselines.
(2) Qualitative analysis with sample prompts and generations for Syngen and the two baselines, demonstrating that SynGen overcomes failure cases of the other two models.
(3) Ablations on the loss components and loss weighting.

**Strengths:**

1. This paper tackles an important and challenging drawback of text-to-image diffusion models struggling to faithfully generate objects with the right attributes.
2. This paper proposes an interesting two-part loss that imposes localized pairwise constraints on the attention maps between image patches and token embeddings. The first constraint enforces that the same image patches attend to both the noun and its modifiers. The second constraint enforces that image patches attending to these nouns/modifiers do not attend to any other tokens in the prompt.
3. This paper attempts to evaluate on a large set of prompts including those from prior work as well as proposing their own challenge set.
4. This paper attempts to better understand the effects and side-effects of the proposed losses via ablations.


**Weaknesses:**

1. The proposed method appears to be strongly grounded in the framework of the Attend-and-Excite paper. The writing doesn’t highlight this connection while introducing the approach in Section 2, beyond a minor footnote. Section 2 would probably need some revision to better draw this connection.
2. Considering that the paper is tackling a very specific problem, the general “content separation” rating collected in the human eval seems pretty weak. Why not collect finer-grained scores for: number of objects in the prompt, number of objects from prompt generated, number of objects from prompt generated with the correct attributes? Wouldn’t this give a much better sense of task success and could potentially allow computing recall?
3. The loss function ablation mentions that sometimes objects are omitted from the generated image. Is combining the proposed losses with global constraint defined in the Attend-and-Excite paper a viable option? Since the framework is identical, wouldn’t combining the losses have mitigated this issue? The Attend-and-Excite paper talks about attribute binding as well, so this appears to be a variant to try.

Nits:
1. Maybe more explicitly state that this is an inference-only guidance procedure?
2. There’s a typo on line 112 for the equation: $\\nabla_{z_t}\\mathcal{L}$ not $\\nabla z_t\\mathcal{L}$. Also $z’_t$ not $z’t$.


**Questions:**

1. Was there a reason to not include numerical modifiers in the study?
2. Are there unexpected side-effects of such constraints on model generalization?

**Limitations:**

One limitation that was perhaps overlooked is the lack of standardized evaluation. Having to rely on different crowdsourcing tasks and qualitative comparisons makes it much harder to track progress.

---

> ### Author Rebuttal · Authors · 2023-08-09
>
> **W1: Writing doesn’t highlight the connection to A&E. Especially section 2:** Thank you for pointing out that the connection to A&E did not come across clearly. We will revise section 2 to give just credit and clarify the relation to A&E.
>
> **W2: "Concept Separation" is too general. Consider collecting finer-grained scores for: number of objects in the prompt, number of objects from its corresponding generated image, number of objects with correct attribute in the corresponding generated image. This provides more sense of success and allows to compute recall.**  Thank you for this important suggestion. Following this suggestion, we collected finer-grained ratings as requested for 120 images for all four baselines in the DVMP dataset. In the table below, the top row shows the fraction of images where raters stated that an entity was missing from the image (sum of number of objects in the image divided by the actual number of objects in the text). The bottom row shows the fraction of images where raters found the attribute was missing or incorrectly bound (sum of number of attribute errors divided by the actual number of attributes in the text). In this dataset, SynGen is on par with A&E in terms of entity neglect, and better in terms of improper binding. We will provide more analysis in the final version.
>
> |                               | SynGen | A&E  | Structured | Standard SD |
> |-------------------------------|--------|------|------------|-------------|
> | Entity Neglect Rate (lower is better) | **20.45** | 22.94 | 31.67     | 33.78      |
> | Improper Binding (lower is better)    | **46.27** | 57.51 | 66.84     | 64.14      |
>
>
>
> **W3: Is combining the proposed losses with A&E’s global constraint a viable option?**
> Yes, we believe that should work well. We note that A&E uses Gaussian Smoothing and ‘Iterative Latent Refinement’ that we find to hurt for our task, but both losses can be used.  We will discuss in the paper together with the response to W1.
>
> **Nits:** We will address the typo noted as well as state more explicitly that no training is needed.
>
>
> **Q1 Why didn’t you include numerical modifiers in the study:**
>
> We agree that this is an interesting future direction for SynGen. Our preliminary work showed that controlling the number of instances of an object has some fundamental differences with the task discussed here. In terms of attention maps, a numerical modifier behaves very differently than modifiers like color, because it induces different properties of the attention maps. For instance, in an image with “two trees” you expect the attention map to usually have two non-overlapping blobs.
> Due to these differences, we decided to leave numerical modifiers for future work.
>
> **Q2: Are there unexpected side-effects of such constraints on model generalization?**
> As stated in the Limitations section, we observe that the visual appeal of images generated degrades with the number of modifiers in the prompt. However, SynGen’s decline is remarkably less pronounced compared to existing models (see figure 12 in the appendix).
>
> **The lack of standardized evaluation makes it difficult to track progress:**
> We agree with the reviewer that this is an important topic.
>
> In fact, we spent significant effort to develop automated evaluation metrics and our best attempt is recorded in section G and table 4 in the appendix. However, multimodal models notoriously fail in groundedness, and human agreement is low. One such example can be seen in the evaluation of the StructureDiffusion paper. There, in table 1, the automatic measure only agrees with human evaluation ~ 47% of the time, where 33% is random. The automatic measure we devise reaches better human agreement (43.5%, where 25% is random), but it is still low. We thus opted to keep it in the appendix, to provide a way of tracking progress, while not over-emphasizing these results.  We believe that given these limitations, it is futile, at the current state of the research, to rely on automatic evaluation in this task. Rather, we opted for high quality and fine-grained human evaluation.
>
> Following this comment, we will discuss the need for automated metrics in the paper, and encourage the community to develop some. We will share the raw data of our experiments so future papers can compute future metrics on our data.

---

> > ### Comment · Reviewer_LWw2 · 2023-08-17
> > **Re: Author response**
> >
> > Thanks for the additional experiments and evaluation! There are lots of details brushed under the rug regarding the full evaluation setup but happy to give the benefit of the doubt that these details will be shared and that the evaluation is robust. I've raised my score to reflect this.

---

> > > ### Author Response · Authors · 2023-08-17
> > >
> > > Thank you so much for the support and trust. We will provide all details of the experiments and make our code public to make experiments easy to replicate.

---

### Official Review · Reviewer_EHJQ · 2023-07-06

**Soundness:** 4 excellent
**Presentation:** 4 excellent
**Contribution:** 4 excellent
**Rating:** 8
**Confidence:** 3

**Summary:**

A frequent issue in text-prompted image generation (and in many other grounded language scenarios) is that a model will treat text akin to a bag of words and ignore syntactic relationships, such as which adjective attaches to which noun -- this is referred to as a problem with lingusitic binding. The paper proposes addressing this problem in diffusion models by adding an extra step to the diffusion process that nudges the image in the process of being generated so as to respect modifier relationships extracted from a syntactic parse of the prompt. This intervention operates in terms of cross-attention maps (i.e. attention relationships established between tokens and parts of the image), with the intuition that related tokens should map to the same region of the image, and unrelated tokens to different regions.

The paper also introduces a new challenge dataset aimed at diagnosing problems with linguistic binding in image generation (the proposed method performs well on this challenge set).

**Strengths:**

The issue of improper linguistic binding is one that plagues not only attempts at image generation, but many other model uses of grounded language. It's an important problem that has not been adequately resolved by prior work, whereas this paper has demonstrated substantial steps to overcoming this issue without sacrificing on generated image quality. All of this speaks to the significance of this work, which has the potential to see broad and immediate application without the need to undertake costly new model training efforts.

The paper is clearly written, and makes effective use of figures to illustrate the problem and how the proposed method resolves it.

There is thorough and convincing human evaluation on both existing data, and a new challenge dataset designed specifically to reveal instances of problems with binding.

**Weaknesses:**

There are not many weaknesses to point to. This is not a weakness of the paper per se, but when the method my main remaining (aesthetic) dissatisfaction is the reliance on an external parser, and the possibility of cascading failure that any such pipeline-based system entails.

**Questions:**

none

**Limitations:**

Limitations are well addressed in the paper.

---

> ### Author Rebuttal · Authors · 2023-08-09
>
> Thank you for your supportive review of this work. The reviewer raises an important point in designing future systems using SynGen. Future work will need to take into account failures of the parser and develop ways to handle them.

---

> > ### Comment · Reviewer_EHJQ · 2023-08-19
> >
> > I have read the other reviews, and would like to thank the authors for their responses.
> >
> > With this information in mind, I will maintain my score.
> >
> > Some questions have been raised regarding similarity to the Attend-and-Excite paper. My view on this is that Attend-and-Excite is different in that it doesn't address the attribute binding problem (except indirectly by decreasing attribute neglect). Attribute binding seems to be somewhat resistant to being solved by indirect methods, which contributes to my favorable assessment of the more direct approach taken in this paper.

---

### Official Review · Reviewer_RH2g · 2023-07-07

**Soundness:** 3 good
**Presentation:** 3 good
**Contribution:** 3 good
**Rating:** 7
**Confidence:** 4

**Summary:**

The paper focuses on attribute binding/leak/neglect problems in text-to-image models. The authors propose, SynGen, a method that utilizes dependency trees combined with cross-attention map optimization to achieve better attribute binding results. Extensive experiments are conducted to show the effectiveness of SynGen compared to previous methods.

**Strengths:**

- The paper addresses an important compositional problem in text-to-image generation.

- The method is intuitive and effective. SynGen can obtain stronger attribute-object association using dependency trees compared to Structure Diffusion, the positive loss design is well-motivated for solving the problem, and the negative loss design enforces the cross-attention maps over A&E.

- The experiment looks comprehensive by considering all sources of datasets and including even more challenging prompts DVMP. The prompts in DVMP are more challenging and realistic than previous "A and B" format prompts.

**Weaknesses:**

- I am concerned with the efficiency of incorporating so many negative losses in the diffusion process. What's the speed of SynGen compared to the original SD?

- Using tree-based methods for binding may fail for more complicated and practical prompts. In reality, SD users write much longer prompts that describe the components at different levels. However, I think the contribution of SynGen still matters a lot as the community needs time to develop from methods for short prompts to more generalized methods.


**Questions:**

Apart from the question in weaknesses:

1. Would SynGen still work for prompts like "an apple that is blue" or "a red apple on the left and another on the right"?

2. I may have missed this part but how do you deal with the padding tokens?

3. Table 2 shows that positive only and positive+negative have lower visual appeal percentages than negative only. Does that imply that the positive loss will harm the visual appealing of the images?

**Limitations:**

See weaknesses.

---

> ### Author Rebuttal · Authors · 2023-08-09
>
> **W1: What is the speed SynGen compared to the original SD?** Syngen is about the same speed as A&E (~10% slower). This is about twice slower than vanilla SD. We did not invest in performance tuning, so we assume speed can be improved considerably once we do. We will add this information to the Limitations and appendix.
>
> **W2: "Using tree-based methods for binding may fail.... However, the contribution of SynGen matters a lot as the community … develops more generalized methods.":**  We agree. Importantly, once better binding is available, it can be easily integrated into the generation process using our SynGen approach.
>
>
> **Q1:Would SynGen work for prompts like "an apple that is blue" or "a red apple on the left and another on the right"?** (1) “An apple that is blue” was not supported in the submitted code, but we already have it working well in the most recent version. Same for “An apple that’s extremely blue” and “An apple that is red and yellow in appearance”. (2) “A red apple on the left and another on the right”: Unfortunately not. There are two issues here. First, regarding "left of”: Spatial relations are not well handled by SD, and SynGen is not designed to fix that problem. Several recent papers proposed ways to improve spatial relations in SD and we assume that combining them with SynGen may help. Second, regarding "another”: SynGen will attribute ‘red’ to ‘apple’, but ‘another’ is an implicit mention of a second red apple, which requires commonsense to identify, and is beyond the scope of our work. More generally, we work with a dependency graph, a syntactic scheme which captures some useful aspects of the structure of the prompt, but is limited to syntactic relations. More elaborate semantic graph schemes can be easily incorporated in our pipeline in the future.
>
> **Q2: How do you deal with the padding tokens?** We only extract the cross-attention maps of tokens that are in the prompt. That is, we ignore the start, end, and padding tokens.
>
> **Q3: Does Table 2 imply that positive loss harms visual appeal of images?** The table demonstrates that to address the binding problem we need both, the positive and the negative loss. If we were to only use the negative term, the images would have been more visually appealing, but like the table shows, it would not address improper binding.

---

> > ### Comment · Reviewer_RH2g · 2023-08-14
> > **Thank you**
> >
> > Thank the authors for addressing my questions. I would like to maintain my score for now.

---

### Official Review · Reviewer_GWNd · 2023-07-09

**Soundness:** 3 good
**Presentation:** 3 good
**Contribution:** 3 good
**Rating:** 6
**Confidence:** 3

**Summary:**

This paper approaches the binding problem in text to image diffusion models that is marked by the inability of the models to appropriately identify which modifiers are attached to which nouns in the input text. This often results either in images that mix up the attributes of the different nouns that are mentioned in the text or in images that completely disregard some of the attributes or fall back to statistically likely combinations that are not mentioned in the text. The paper proposes an inference time optimization based on first identifying which modifiers are associated with which nouns, and then optimizing a loss function that explicitly enforces the cross attention matrices of corresponding (i.e. related by modifier-noun relationship) tokens in the text to be similar, while those of every other pair in the sentence to be dissimilar. This results in a diffusion model that adheres more faithfully to the input text and produces images that are deemed better more often than competing baseline approaches that attack the same problem.

**Strengths:**

## Originality, Quality, Significance
* The paper provides a novel lightweight method using off the shelf syntactic parsers to enforce better binding of modifiers to nouns in text to image diffusion models. The qualitative results seem promising and the human evaluation results suggest that the proposed method outperforms previous approaches to mitigate the binding issue.

## Clarity
The paper is easy to follow. The qualitative examples demonstrate the different kind of issues faced by the baselines and the improvements brought by the proposed approach.

**Weaknesses:**

* Some qualitative examples on the collected DVMP dataset with some categorization on failure cases based on having uncommon modifiers vs different number (performance on 1,2,3 and more modifiers) of modifiers would have been good to see.

**Questions:**

## Clarifications
* The metric used in the human evaluation in Table 1 is not sufficiently clear. Aren't the values in each column supposed to sum to 100?
* What is the Complex Concepts Prompts dataset mentioned in Fig 5?

## Suggestions
* It would be easier for the reader to follow if the examples in Figure 4,5,6 are reorganized such that the references to them in the text are chronological. Currently it requires the reader to keep going back and forth across the figures.

**Limitations:**

Yes

---

> ### Author Rebuttal · Authors · 2023-08-09
>
> **Provide qualitative examples having uncommon modifiers vs different number:** Following this suggestion. We will show qualitative examples of SynGen and our baselines on prompts with a varying number of modifiers. Specifically, with 2, 3, 4, 5, and 6 modifiers. Note that the DVMP dataset does not contain prompts with just a single modifier. In this context, we note that figure 12 in the appendix quantitatively compares SynGen with A&E over several numbers of modifiers in the DVMP dataset.
>
> **Q1: Should values in Table 1 sum to 100%?** Yes. The current sum of ~ 99.8 was due to a rounding error. We’ll fix it in the final version.
>
> **Q2: What is the Complex Concepts Prompts dataset mentioned in Fig 5?**
> This was an earlier name we considered for our DVMP data. Thank you for noting this mistake.
>
> **Suggestions 1: Reorganize Figures 4, 5, 6 for chronological reference in the text:**
> We agree with your suggestion and will rearrange the figures to be congruent with the text describing them. Thank you for helping us improve the paper.

---

> > ### Comment · Reviewer_GWNd · 2023-08-19
> > **Thanks for the clarifications**
> >
> > Thanks, and looking forward to the qualitative examples and categorization of errors! Maintaining the score for now.

---

### Decision · Program_Chairs · 2023-09-21

**Decision:**

Accept (oral)

**Comment:**

This paper proposes an approach to improve diffusion-based image generation models with respect to image–text alignment, in particular, improving the correspondence between entity-noun and its modifier (in the text prompt) and the generated object and its attributes (in the image). The reviewers find the experiments comprehensive and the paper is clear. The concern around the design of the human annotation task is discussed and addressed during the rebuttal, and the authors will provide this additional analysis to the final paper. Overall I agree with the reviewers that the paper addresses an interesting problem in a simple but effective way (which is shown through the experiments).